# ChainedDiffuser: Unifying Trajectory Diffusion and Keypose Prediction for Robotic Manipulation

**Zhou Xian**[*], **Nikolaos Gkanatsios**[*], **Theophile Gervet**[*], **Tsung-Wei Ke, Katerina Fragkiadaki**

School of Computer Science
Carnegie Mellon University
{xianz1, ngkanats, tgervet, tsungwek, katef}@cs.cmu.edu

chained-diffuser.github.io

**Abstract:** We present ChainedDiffuser, a policy architecture that unifies action keypose prediction and trajectory diffusion generation for learning robot manipulation from demonstrations. Our main innovation is to use a global transformer-based action predictor to predict actions at keyframes, a task that requires multi-modal semantic scene understanding, and to use a local trajectory diffuser to predict trajectory segments that connect predicted macro-actions. ChainedDiffuser sets a new record on established manipulation benchmarks, and outperforms both state-of-the-art keypose (macro-action) prediction models that use motion planners for trajectory prediction, and trajectory diffusion policies that do not predict keyframe macro-actions. We conduct experiments in both simulated and real-world environments and demonstrate ChainedDiffuser's ability to solve a wide range of manipulation tasks involving interactions with diverse objects.

**Keywords:** Manipulation, Imitation Learning, Transformers, Diffusion Models

## 1  Introduction

While learning manipulation policies from demonstrations is a supervised learning problem, the multimodality and diversity of action trajectories poses significant challenges to machine learning methods. Some tasks, such as placing a cup in a cabinet, can be handled by a policy that provides only a desired goal pose for the cup [1, 2, 3], while others, such as wiping off dirt on the floor, necessitate the policy to generate a continuous action trajectory [4, 5] for the grasped mop.

One line of manipulation learning methods models action trajectories from demonstrations. These methods either reactively map vision and language to dense temporal actions [6, 7, 8, 5, 9], or model the input-action compatibility using energy-based models [10, 11, 12, 13]. Despite recent progress, these methods may struggle with multimodal action trajectory distributions, or experience training stabilities [14, 13, 15]. Building on successes in diffusion models [16, 17, 18], a recent line of work proposes to train diffusion-based policies [14, 4, 19] for generating action trajectories. These approaches have demonstrated stable training behavior and impressive capability in capturing multimodal action trajectory distributions. Yet, they have not yet been tested on long-horizon manipulation tasks.

Another line of works casts the problem of robot manipulation as predicting a sequence of discrete end-effector actions on keyframes [1, 20, 12, 21]. This paradigm extracts keyframes from continuous demonstrations and predicts end-effector actions in these keyframes [2, 22, 3, 21]. Subsequently, a low-level path planner connects the predicted keyposes (*macro-actions*), and returns full trajectories

---

[*] Equal contribution

7th Conference on Robot Learning (CoRL 2023), Atlanta, USA.

that adhere to both environmental and task constraints. Leveraging recent advances in attention-based architectures [23], a number of methods extend keyframe action prediction to 6-DoF language-instructed manipulation tasks [2, 3, 22, 24, 21].

The assumptions behind keyframe prediction hinder its applicability to manipulation tasks that extend beyond pick-and-place type of actions. Many tasks, such as wiping a table, opening a door while respecting the kinematic constraints, etc., can only be solved via continuous interactions with the environment. Moreover, the dependence on low-level path planning further restricts these methods' capability: while a range of tasks need collision-free trajectories, other tasks, such as object pushing [24, 3, 25], necessitate that the motion planner disregards collision avoidance. Although supervision for this additional reasoning is readily available in simulated datasets [26], real-world human demonstrations typically lack such data, not to mention that collision-free motion planning in the real world requires accurate state estimation, which presents its own challenges.

In light of the above, we present *ChainedDiffuser*, a neural architecture that unifies the two afore-mentioned paradigms. ChainedDiffuser is a policy architecture that takes as input visual signals and, optionally, a language instruction and outputs temporally dense end-effector actions. At a coarse level, it predicts macro-step end-effector actions (which we will call *macro-actions*), a high-level task that requires global comprehension of the visual environment and the task to complete, with a global transformer-based action predictor. Then, a low-level trajectory diffuser generates local trajectory segments to connect the predicted macro-actions. In comparison to transformer-based macro-step prediction methods [2, 3, 22, 24], our model predicts smooth trajectories to accommodate tasks that require continuous interactions and collision-free actions. In comparison to diffusion-only trajectory generation methods [14, 15, 4, 19], our hierarchical approach handles long-horizon tasks in a more structured manner and allows different modules to concentrate on the tasks at which they excel.

We test ChainedDiffuser on RLBench [26], an established benchmark for manipulation learning from demonstrations. We evaluate our model across a variety of tasks and scenarios studied in previous literature [22, 24]. ChainedDiffuser sets a new state of the art, and outperforms ablative versions that do not predict macro-actions or use regression or motion planners for keyframe-to-keyframe trajectory prediction. Furthermore, we validate our model in real-world scenarios with a number of long-horizon manipulation tasks, using a handful of human demonstrations for training.

## 2 Related Work

**Learning from Demonstrations** [27, 28] has been a common paradigm for robotics but requires demonstration data collection in the real world [6, 29, 30] or simulation [26, 31, 32]. To improve data efficiency, several approaches learn the policy on top of pre-trained visual representations that exploit large vision-only datasets [33, 34, 35, 36, 37, 38]. Orthogonal to this, other approaches abstract every task as a sequence of subgoals, expressed as pick-and-place primitives [1, 2] or keyframes [39, 40]. In this case, hand-designed low-level controllers are employed to plan the end-effector's motion between intermediate subgoals. While data-efficient, this abstraction does not generalize adequately to scenarios where only few specific trajectories that respect all physical constraints are valid [41], such as manipulations of deformable [42, 43] or articulated [44] objects, motions of closed-chain robotic systems [45, 46], or trajectories through obstacles in a cluttered environment [47]. As a result, recent works resort to semi-manual cost specification for each additional constraint (e.g., collision avoidance, trajectory smoothness [4]). Closer to our approach, James and Abbeel [41] learn to score trajectories proposed by either hand-designed or learning-based planners. Instead, we train scene conditioned diffusion models to generate trajectories that connect predicted keyposes.

**Transformers for Robotics** Following their success in natural language processing [23, 48, 49] and computer vision [50, 51], numerous recent works use Transformer-based architectures for robotics and control [52, 53, 6, 54, 55, 21]. One main motivation is the flexibility of attention for long-horizon prediction when combining information from multiple sensory streams, such as visual observations and language instructions [56, 22]. Most related to ours is the stream of multi-tasking Transformer-

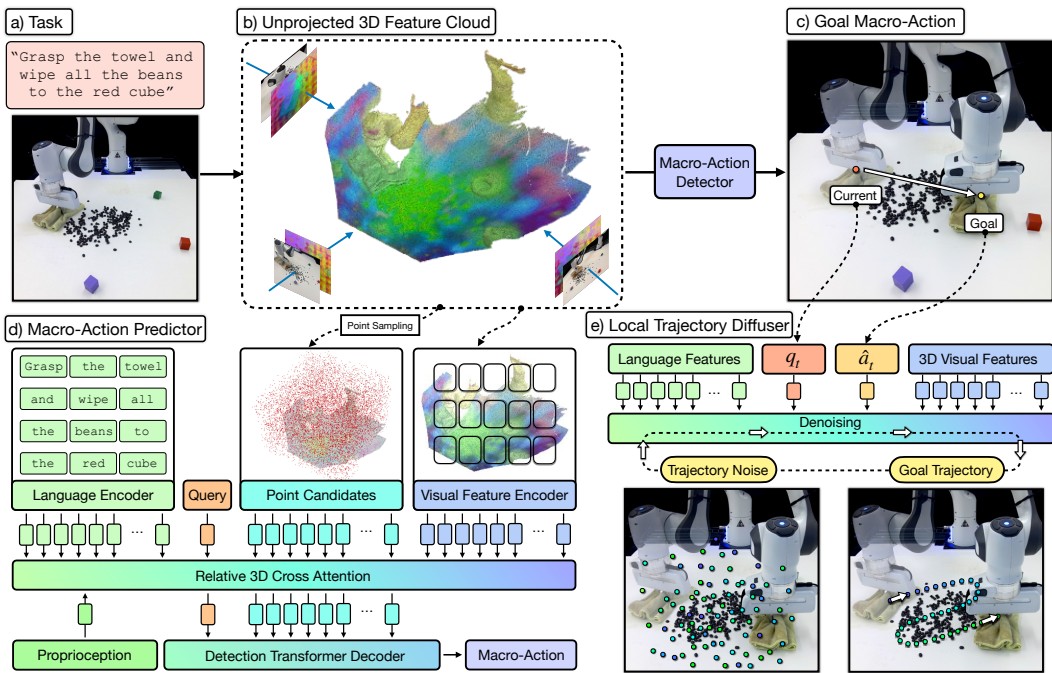

Figure 1: **ChainedDiffuser** is a robot manipulation policy architecture that predicts a set of robot keyposes and links them using predicted trajectory segments. It featurizes input multi-view images using pre-trained 2D image backbones and lifts the resulting 2D feature maps to 3D using sensed depth. In (b) we visualize the 3D feature cloud using PCA and keeping the 3 principal components, mapping them to RGB. The model then predicts end-effector keyposes using coarse-to-fine attention operations to estimate a 3D action map for the end-effector's 3D location and regress the robot's 3D orientation, similar to [21] (d). It then links the current end-effector pose to the predicted one with a trajectory predicted using a diffusion model conditioned on the 3D scene feature cloud and predicted keypose (e).

based models, that are trained on diverse datasets to achieve higher in-distribution [3, 24] or out-of-distribution generalization [6, 57, 58, 59]. Our model comprises of two attention-based modules, one for macro-step action prediction and one for local trajectory optimization, that can leverage different input modalities and operate over different abstractions.

**Diffusion Models** [60, 18, 16, 17] learn to approximate the data distribution through an iterative denoising process, and have shown impressive results on both unconditional and conditional image generation [61, 62, 63, 64]. In the field of robotics, diffusion models find applications on planning [15, 65, 66], scene re-arrangement [67, 68], controllable motion optimization [69, 70], video generation [71] and imitation learning [14, 19]. Their main advantage is that they can better capture the action trajectory distribution compared to previous generative models. Recent works use diffusion model to predict complete trajectories, often auto-regressively [14, 72]. Instead, we use diffusion models to generate local trajectories that are chained together with macro-actions.

## 3 ChainedDiffuser

### 3.1 Overview

The architecture of ChainedDiffuser is illustrated in Figure 1. ChainedDiffuser combines macro-action prediction with conditional trajectory diffusion. Its input comprises of visual observations of the environment and a natural language description $l$ of the task. At each step, ChainedDiffuser predicts a macro-action $\hat{a}_t$ using a global policy $\pi_{\text{global}}$, and then feeds $\hat{a}_t$ together with its current end-effector state $q_t$ to a low-level local trajectory generator $\pi_{\text{local}}(q_t, \hat{a}_t)$ to generate dense micro-actions connecting $q_t$ and $\hat{a}_t$, as shown in Figure 1. Both $a_t$ and $q_t$ share the same space $\mathcal{A} =$

$\{a_{\text{pos}}, a_{\text{rot}}, a_{\text{grip}}\}$, consisting of the end-effector's 3D position $a_{\text{pos}}$, rotation $a_{\text{rot}}$ represented as a 4D quaternion, and a binary flag $a_{\text{grip}}$ indicating whether the gripper is open. For each task, we assume access to a dataset $\mathcal{D} = \{\zeta_1, \zeta_2, ..., \zeta_m\}$ of $m$ expert demonstrations, where $\zeta_i$ contains the language instructions $l$, visual observations $o$ and end-effector states $q_t$ for all timesteps in the demonstration.

**Input Encoding**  ChainedDiffuser operates in a 3D space to achieve robustness across changing camera viewpoints – an important advantage over prior 2D methods which assume fixed camera viewpoints [22, 24, 14]. Compared to prior robotic architectures which rely on voxel-based 3D representation (e.g., [3, 40]), ChainedDiffuser employs a point-based representation, that facilitates sparse computation and circumvents precision loss during voxelization. ChainedDiffuser uses a frozen CLIP [73] to encode both the language instruction $l$ and the RGB images $o_t$ into a set of language and visual feature tokens respectively. Then, it uses the depth channel information to unproject the 2D image feature tokens into a 3D feature cloud (Figure 1(b)), where each visual token has 2D appearance information and 3D positional information. We also encode the proprioception information $q_t$ with a simple MLP.

## 3.2 Macro-Action Predictor

Our macro-action predictor $\pi_{\text{global}}$ is based on Act3D [21], a state-of-the-art macro-action prediction method that uses a point-based transformer that casts end-effector action prediction as 3D action map prediction. We include its main pipeline here for completeness. Act3D samples iteratively 3D point candidates and featurizes them using relative position attentions to a scene 3D feature cloud. Then, a trainable query token $\mathcal{Z}_{\text{query}}$ is used to score a pool of $N$ point candidates $\{P_i = \langle x_i, y_i, z_i \rangle\}_{i=1}^N$ in the scene and select a position for next macro-action. The point candidates are first uniformly sampled within the robot's empty workspace and only contain 3D positional information and a trainable feature embedding $\mathcal{Z}_{\text{point}}$. The query token and the point candidates individually attend to the concatenation (across the sequence dimension) of language tokens $\mathcal{Z}_{\text{ins}}$, visual feature tokens $\mathcal{Z}_{\text{vis}}$ and proprioception token $\mathcal{Z}_{\text{robot}}$ (Figure 1(d)):

$$\tilde{\mathcal{Z}}_{\text{query}} = \text{Attn}\Big(\mathcal{Z}_{\text{query}}, \langle \mathcal{Z}_{\text{ins}}, \mathcal{Z}_{\text{vis}}, \mathcal{Z}_{\text{robot}} \rangle\Big) \tag{1}$$

$$\tilde{\mathcal{Z}}_{\text{point}} = \text{Attn}\Big(\mathcal{Z}_{\text{point}}, \langle \mathcal{Z}_{\text{ins}}, \mathcal{Z}_{\text{vis}}, \mathcal{Z}_{\text{robot}} \rangle\Big) \tag{2}$$

where $\text{Attn}(x, y)$ is an attention operation [23, 74] where the queries are formed from $x$, the keys and values from $y$. After this contextualization step, the query token and the point candidates have captured the task and scene information. We take the dot product of the contextualized query embedding with all point candidates and select the best-matching point candidate for the position of the predicted macro-action:

$$\hat{a}_{\text{pos}} = \langle x_{\hat{i}}, y_{\hat{i}}, z_{\hat{i}} \rangle, \quad \hat{i} = \underset{i}{\arg\max}\, \tilde{\mathcal{Z}}_{\text{query}}^T \cdot \tilde{\mathcal{Z}}_{\text{point}}^i \tag{3}$$

Once we obtain the best point candidate, we predict the rotation and gripper open flag with a simple MLP on top of the query:

$$\langle \hat{a}_{\text{rot}}, \hat{a}_{\text{grip}} \rangle = \text{MLP}(\tilde{\mathcal{Z}}_{\text{query}}) \tag{4}$$

## 3.3 Local Trajectory Diffuser

Once we obtain the macro-action $\hat{a}_t$ for the current step $t$, we call upon our diffusion-based local trajectory generator to fill up the gap in-between with micro-actions. We model such trajectory generation as a denoising process [18, 14, 4]: we start with drawing a sequence of $S$ random Gaussian samples $\{\mathbf{x}_s^K\}_{s=1}^S$ in the normalized SE(3) space, and then perform $K$ denoising iterations to transform the noisy trajectories to a sequence of noise-free waypoints $\{\mathbf{x}_s^0\}_{s=1}^S$. Each denoising iteration is described by:

$$\mathbf{x}_s^{k-1} = \lambda_k(\mathbf{x}_s^k - \gamma_k \epsilon_\theta(\mathbf{x}_s^k, k)) + \mathcal{N}(0, \sigma_k^2 I), \quad 1 \leq s \leq S \tag{5}$$

where $\epsilon_\theta$ is the noise prediction network, $k$ the denoising step, $\mathcal{N}(0, \sigma_k^2 I)$ the Gaussian noise added at each iteration, and $\lambda_k, \gamma_k, \sigma_k$ are scalar noise schedule functions dependent on $k$ (Appendix 7.1).

The noise prediction network (Figure 1 (e)) is also an attention-based model that absorbs similar input as the macro-action selector does, i.e., the language instruction $l$, RGB-D observations $o_t$ and current end-effector state $q_t$, but additionally conditions on the goal macro-action $\hat{a}_t$ and the denoising timestep $k$. The language tokens $\mathcal{Z}_{\text{ins}}$, visual tokens $\mathcal{Z}_{\text{vis}}$ and current end-effector state $\mathcal{Z}_{\text{robot}}$ are featurized similarly to the Macro-Action Selector. We use an MLP to encode the goal macro-action into $\mathcal{Z}_{\text{macro}} = \text{MLP}(\hat{a}_t)$. We encode the denoising timestep into $\mathcal{Z}_{\text{time}}$ using sinusoidal positional embeddings [23], and encode the the sampled noise using an MLP into a sequence of tokens $\mathcal{Z}_s^k$. We let this sequence iteratively cross-attend to all encoded inputs first:

$$\tilde{\mathcal{Z}}_s^k = \text{Attn}(\mathcal{Z}_s^k, \langle \mathcal{Z}_{\text{ins}}, \mathcal{Z}_{\text{vis}}, \mathcal{Z}_{\text{robot}}, \mathcal{Z}_{\text{macro}}, \mathcal{Z}_{\text{time}} \rangle),$$

and then self-attend to obtain a finalized $\tilde{\mathcal{Z}}_s^k$ (note that we reuse the same symbol for presentation clarity):

$$\tilde{\mathcal{Z}}_s^k = \text{Attn}(\tilde{\mathcal{Z}}_s^k, \tilde{\mathcal{Z}}_s^k)$$

Again, we use relative positional embeddings to encode all tokens' spatial positions. For the trajectory noise tokens, we additionally encode each sample's temporal position $s$ using sinusoidal positional embeddings. These are added to the respective noise tokens $\mathcal{Z}_s^k$. The contextualized noise sample is then fed into another MLP for noise regression:

$$\epsilon_\theta(\mathbf{x}_s^k, k) = \text{MLP}(\tilde{\mathcal{Z}}_s^k) \tag{6}$$

After $K$ denoising steps by substituting Equation 6 into 5, we convert the denoised samples back to the actual micro-actions by unnormalizing them: $a_{t-1+s} = \text{Unnormalize}(\mathbf{x}_s^0), \quad 1 \le s \le S$. For more implementation and training details, please see the Appendix 3.4.

**Noise schedulers** We model local trajectory optimization as a discrete-time diffusion process, which we implement using the DDPM sampler [18]. DDPM uses a non-parametric time-dependent noise variance scheduler $\beta_k$, which defines how much noise is added at each time step. We adopt a scaled linear schedule for the position and a squared cosine schedule for the rotation of each trajectory step.

### 3.4 Implementation and Training Details

ChainedDiffuser takes as input $m$ multi-view RGB-D images of the scene. For experiments in simulation, we use $m = 3$ (*left*, *right*, *wrist*) or $m = 4$ (with an additional *front* view), depending on the settings of the baselines we compare with. For real-world experiments, we use $k = 1$, with a single front-view camera. Each RGB-D image is $256 \times 256$ and is encoded to $64 \times 64$ visual tokens with CLIP's ResNet50 visual encoder [73]. The demonstration data contains end-effector states for all timesteps. In order to extract macro-actions to supervise the action selection transformer, we use a simple heuristic following previous literature [3, 22, 24]: a timestep is considered to be a keyframe containing macro-action if the gripper opens or closes, or if the robot arm is not moving (when all joint velocities approach zero). All dense actions present in the demonstration are used to supervise the local trajectory diffuser. We resample the dense trajectories between extracted macro-actions to a trajectory of fixed length $S = 50$. We found in practice, denoising fixed number of micro-actions leads to more stable training, and works better than learning variable-length trajectory diffusion with predicted trajectory length. We train both the action detector and the trajectory diffuser jointly, using a cross-entropy (CE) loss to supervise the point candidate selection by predicting a probability distribution $q$ over all point candidates in the pool, and mean-sqaured error (MSE) losses to supervise quaternion, gripper opening and trajectory noise regression:

$$\mathcal{L} = \frac{1}{|\mathcal{D}||\zeta|} \sum_{\zeta \in \mathcal{D}} \sum_{\hat{t} \in \zeta} \left[ \text{CE}(q(\{P_i\}^N), q^*(\{P_i\}^N)) + \text{MSE}(\hat{a}_{\hat{t}}, \hat{a}_{\hat{t}}^*) + \sum_{t=\hat{t}}^{\hat{t}+S-1} \text{MSE}(\epsilon_\theta(\mathbf{x}_s^k, k), \epsilon_k) \right],$$
$$\tag{7}$$

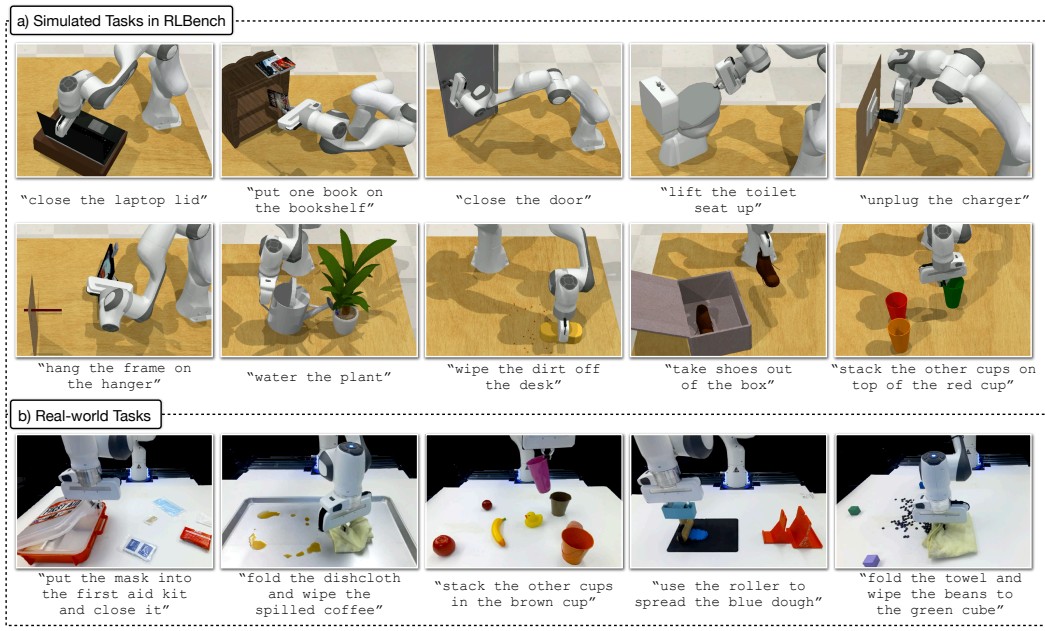

Figure 2: Simulation and real-world tasks we evaluate on.

where $*$ indicates predicted value, $\hat{a}_{\hat{t}} = \langle \hat{a}_{\text{rot}}, \hat{a}_{\text{grip}} \rangle$, $k$ is a randomly sampled denoising step, and $\epsilon_k$ is the sampled ground truth noise. In order to speed up training in practice, we train the first 2 terms till convergence, and then add the 3rd term for joint optimization, as opposed to using ground-truth macro-actions for training the trajectory diffuser. This allows the trajectory diffuser to incorporate certain error recovery capability to handle inaccurate macro-action predictions. In addition, at test time, we normalize the predicted quaternion to ensure it respects the normalization constraint before feeding it to the robot. We use a batch size of $B = 24$ and AdamW [75] optimizer with a learning rate of $1e - 4$ for all our experiments. Our single-task model is trained for 1 day on one A100 GPU, and multi-task model is trained for 5 days on 4 A100 GPUs.

**Control** Our control algorithm is closed-loop at the macro-action level, which means the macro-action predictor reasons about the surroundings and predicts actions to handle environment changes. At the low-level, our controller is open-loop and follows a Cartesian-space end-effector trajectory composed of the predicted micro-actions using position control, with a control frequency of 10Hz. For our real-robot setup, we use the open source `frankapy` package [76] with ROS, which uses a low-level PID controller at 1kHz.

## 4 Experiments

We test ChainedDiffuser in various manipulation tasks in both simulated and real-world environments. Our experiments aim to answer the following questions:

- How does ChainedDiffuser compare to previous SOTA 2D and 3D manipulation methods?
- Is macro-action prediction helpful in guiding trajectory generation?
- Does ChainedDiffuser work in the real-world where only a single camera and limited number of demonstrations are available?

### 4.1 Simulation Experiments

We run experiments in simulation using RLBench [26], a widely adopted manipulation benchmark with diverse tasks concerning interactions with a wide range of objects, as shown in Figure 2. We

Table 1: Success rates in 10 single-tasks of the Hiveformer experimental setting.

| | pick & lift | pick-up cup | push button | put knife | put money | reach target | slide block | stack wine | take money | take umbrella | Mean |
|---|---|---|---|---|---|---|---|---|---|---|---|
| Auto-λ [77] | 87 | 78 | 95 | 31 | 62 | 100 | 36 | 23 | 38 | 37 | 55.0 |
| HiveFormer [24] | 92 | 77 | 100 | 70 | 96 | 100 | 95 | 82 | 82 | 90 | 88.4 |
| InstructRL [22] | 98 | 85 | **100** | 85 | **99** | 100 | **98** | **93** | 90 | 93 | 93.8 |
| ChainedDiffuser (ours) | **98** | **94** | 96 | **91** | 98 | **100** | 95 | 90 | **100** | **96** | **95.8** |

follow the same setting used in prior works [22, 24], where each task has multiple variations and contains 100 demonstrations. We report success rates in each task averaged over 100 unseen test episodes. For baselines, when possible, we use the official numbers reported in their papers.

**Baselines** We compare ChainedDiffuser with the following baselines:

1. Auto-λ [77] and HiveFormer [24], policy learners that operate on multi-view 2.5D images and predict actions by offseting detected points in the input images.

2. *InstructRL* [22], a policy that operates on multi-view 2D images with pre-trained vision and language encoders, and directly predicts 6-DoF end-effector actions.

3. *Act3D* [21], a policy that predicts keyframe end-effector macro-actions with a 3D action detection transformer and relies on low-level motion planner to connect macro-actions.

4. *Open-loop trajectory diffusion*, which is ChainedDiffuser without the macro-action detector, making it a trajectory diffusion model.

5. *Act3D+ trajectory regression*, which replaces the local trajectory diffuser in ChainedDiffuser with a deterministic trajectory regression

**Dataset** We consider the following single-task experimental settings:

- 10 tasks considered in the Auto-λ [77] experimental setup. These tasks are considered by many prior works and this allows us to compare our performance with them.

- 10 tasks in RLbench we identify to require continuous interaction with the environment, such as `wipe_desk` where a wiping trajectory is needed to remove the dirt from a desk, and `open_fridge` where a local trajectory needs to adhere to the kinematic constraint when the robot is grasping the door handle. Most tasks in RLBench can be reasonably solved with only macro-action prediction and motion planners. This set of tasks we consider highlights the limitation of these approaches.

**Results** We train single-task ChainedDiffuser and the baselines. For *Auto-λ*, *HiveFormer* and *InstructRL* we use the numbers reported in the corresponding papers. We show quantitative results in Tables 1 and 2. ChainedDiffuser consistently achieves better performance than prior methods on all task categories. On the set of challenging tasks for motion planners, ChainedDiffuser gives a significant boost of 60% on average. ChainedDiffuser improves upon open-loop trajectory diffusion model, which demonstrates that delegating global macro-action prediction to a high-level policy to guide local trajectory diffusion helps. *Act3D+ trajectory regression* struggles where multi-modal trajectories are present in demonstrations, e.g. `cup_in_cabinet` where multimodal trajectories exist for grasping the cup and feeding into the cabinet in the training set. This demonstrates that modeling trajectory generation as a multi-step denoising process is advantageous over regression-based model, which aligns with conclusions from previous literature [14].

## 4.2 Real-world Experiments

We conduct experiments with a real-world setup, using a Franka Emika Panda robot with a parallel-jaw gripper. We use a *single* Azure Kinect camera to collect front-view RGB-D image input. See Appendix 7.4 for more details on our hardware and data collection setup. We design 7 tasks that involve

Table 2: Success rates on challenging tasks for motion planners.

| | unplug charger | close door | open box | open fridge | frame off hanger | open oven | books on shelf | wipe desk | cup in cabinet | shoes out of box | Mean |
|---|---|---|---|---|---|---|---|---|---|---|---|
| Act3D [21] | 48 | 9 | 9 | 19 | 66 | 2 | 34 | 4 | 0 | 19 | 21.0 |
| Open-loop trajectory diffusion | 65 | 21 | 46 | 37 | 43 | 16 | 40 | 34 | 6 | 9 | 31.7 |
| Act3D + trajectory regression | 95 | 5 | 95 | 60 | 77 | 17 | 68 | **70** | 40 | 67 | 59.6 |
| ChainedDiffuser (ours) | **95** | **76** | **96** | **68** | **85** | **86** | **92** | 65 | **68** | **78** | **80.9** |

multi-step actions and continuous interactions with the scene (5 are shown in Figure 2), collected $10 - 20$ demos for each tasks, and train a multi-task ChainedDiffuser for real-world deployment.

We refer the reader to our supplementary video for qualitative executions of the robot. We evaluate it on 10 episodes for each task, and report success rates in Table 3. ChainedDiffuser is able to perform reasonably well on most of the tasks, even for tasks with multiple action modes and skills. The most common failure case is caused by noisy depth image: we leverage point selection for macro-action prediction, which would suffer from incorrect depth estimation in the real world. This could potentially be resolved by more accurate camera calibration with a multi-view camera setup and learning to recover from noisy input, which we leave as our future work.

| Task | # Train | Success |
|---|---|---|
| put mask in kit | 20 | 6/10 |
| fold and wipe coffee | 20 | 8/10 |
| stack cups | 15 | 7/10 |
| spread dough | 15 | 7/10 |
| fold and wip beans | 20 | 6/10 |
| put nails in box | 20 | 6/10 |
| press stapler | 10 | 10/10 |

Table 3: Real-world tasks.

### 4.3 Limitations

Our method currently has the following limitations: **1)** Our trajectory diffuser is conditioned on end-effector poses in SE(3) space. It would be ideal to extend it to full joint configuration space for more flexible trajectory prediction. **2)** Our model performs closed-loop control on the macro-action level, which restricts its flexibility in highly dynamic environments. That said, our framework can be easily extended with closed-loop re-planning at the micro-action level, making the policy more robust to environment dynamics, which we leave as our future work. **3)** Following the standard setting in RLBench, our method assumes access to calibrated cameras. We believe this assumption is valid as mobile robots performing household tasks for humans in the future should have cameras attached to the robots, where these cameras can be calibrated when coming out of the factories.

## 5 Conclusion

We presented ChainedDiffuser, a neural policy architecture for learning 6-DoF robot manipulation from demonstrations. Our model achieves competitive performance on various task settings, in both simulation and the real-world. Our experiments demonstrate that by unifying both transformer-based macro-action detection and diffusion-based trajectory generation, ChainedDiffuser achieves the best of both families and addresses their respective limitations. ChainedDiffuser outperforms both keyframe prediction methods and trajectory diffusion alone, which justifies their unification in our framework. It sets a new state-of-the-art in RLbench, and especially improves performance on contact-rich tasks and tasks that involve articulated objects, where methods that rely on hand designed planners typically struggle.

## 6 Acknowledgements

This work is supported by Sony AI, NSF award No 1849287, DARPA Machine Common Sense, an Amazon faculty award, and an NSF CAREER award.

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

# 7 Appendix

## 7.1 Noise schedulers for Local Trajectory Diffuser

We model local trajectory optimization as a discrete-time diffusion process, which we implement using the DDPM sampler [18]. DDPM uses a non-parametric time-dependent noise variance scheduler $\beta_k$, which defines how much noise is added at each time step. We adopt a scaled linear schedule for the position and a squared cosine schedule for the rotation of each trajectory step.

$$x_{k-1} = \frac{\sqrt{\bar{\alpha}_{k-1}}\beta_k}{1-\bar{\alpha}_k}(x_k - \epsilon_\theta(x_k, k,)) + \frac{\sqrt{\alpha_k}(1-\bar{\alpha}_{k-1})}{1-\bar{\alpha}_k}x_k + \frac{1-\bar{\alpha}_{k-1}}{1-\bar{\alpha}_k}\beta_k \mathbf{z} \tag{8}$$

By defining $\alpha_k = 1 - \beta_k$, and $\bar{\alpha}_k = \prod_{i=1}^{k}\alpha_i$, we can now obtain the analytical form of $\lambda_k, \gamma_k, \sigma_k$ in Equation 5 as follows:

$$\lambda_k = \frac{1}{\sqrt{\alpha_k}} \tag{9}$$

$$\gamma_k = \frac{1-\alpha_k}{\sqrt{1-\bar{\alpha}_k}} \tag{10}$$

$$\sigma_k = \frac{1-\bar{\alpha}_{k+1}}{1-\bar{\alpha}_k}\beta_k \tag{11}$$

where $k$ is the diffusion denoising timestep.

## 7.2 Act3D Background and Implementation Details

Act3D is a language-conditioned end-effector 6-DoF keypose predictor that learns 3D perceptual representations of arbitrary spatial resolution via recurrent coarse-to-fine 3D point sampling and featurization. Act3D featurizes multi-view RGB images with a pre-trained 2D backbone and lifts them in 3D using depth to obtain a multi-scale 3D scene feature cloud. It then iteratively predicts 3D foci of attention in the empty 3D workspace, samples 3D point grids in their vicinity, and featurizes the sampled 3D points using relative cross-attention to the physical scene feature cloud, language tokens, and proprioception. Act3D detects the 3D point that corresponds to the next best end-effector position using a detection Transformer head, and regresses the rotation, end-effector opening, and planner collision avoidance from the decoder's parametric query.

We extract two feature maps per $256 \times 256$ input image view: $32 \times 32$ coarse visual tokens and $64 \times 64$ fine visual tokens. We use three ghost point sampling stages: first across the entire workspace (roughly 1 meter cube), then in a 16 centimeter diameter ball, and finally in a 4 centimeter diameter ball. The coarsest ghost points attend to a global coarse scene feature cloud ($32 \times 32 \times n_{\text{cam}}$ coarse visual tokens) whereas finer ghost points attend to a local fine scene feature cloud (the closest $32 \times 32 \times n_{\text{cam}}$ out of the total $64 \times 64 \times m_{\text{cam}}$ fine visual tokens). During training, we sample 1000 ghost points in total split equally across the three stages. At inference time, we trade-off extra performance for additional compute by sampling more ghost points than the model ever saw at training time ($20,000$). We use 2 layers of cross-attention and an embedding size 60 for single-task experiments and 120 for multi-task experiments. Training samples are augmented with random crops of RGB-D images and $\pm 45.0$ yaw rotation perturbations (only in the real world as this degrades performance in simulation).

## 7.3 Simulation Setup in RLBench

The RLBench simulation environment uses a Franka Panda robotic arm on a table-top setting. We consider $m = 4$ camera inputs: *left_shoulder*, *right_shoulder*, *wrist*, and *front*, as shown in Figure 3. The *wrist* camera is attached to the robot's end-effector and moves together with the robot. The other 3 are static. To ensure a fair comparison, when comparing with PerAct, we use all 4 cameras following PerAct setting, and use the first 3 cameras when compared with other baselines.

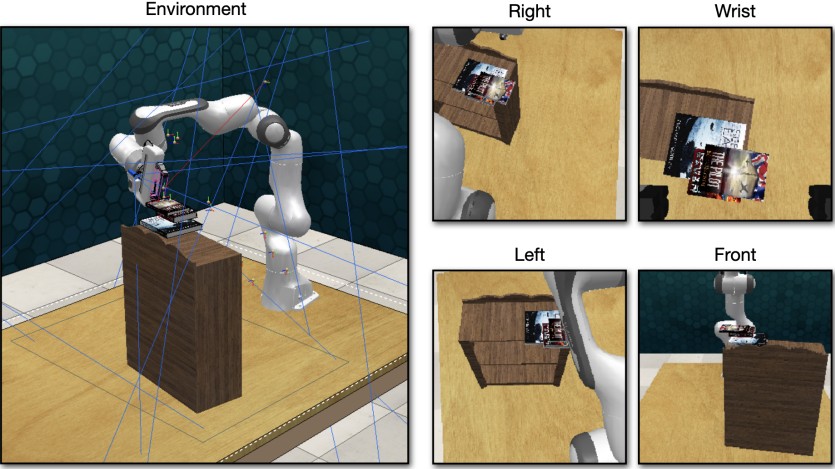

Figure 3: Simulation setup.

## 7.4 Real-world Setup

Our real-robot setup contains a real Franka Panda robotic arm equipped with a parallel jaw gripper, as shown in Figure 4. We use a single Azure Kinect sensor to provide RGB-D input signal from the front view at 30Hz. The image input is of resolution $1280 \times 720$, and we crop and downsample it to $256 \times 256$ before feeding it to our model. We calibrate the extrinsics of the camera with respect to the robot base using the easy_handeye[1] ROS package. Our full model generates dense trajectories, thus we do not use low-level motion planners. We collect 6-DoF human demonstrations by tele-operating the robot using a SpaceMouse[2] at 30Hz, following [14]. We use the same strategy for keyframe extraction as in simulation. Our real-world multi-task policy is trained on 4 A100 GPUs for 3 days. Inference is done on a desktop with a single RTX4090 GPU, running Ubuntu 20.04 and ROS Noetic. For robot control, we use the open-source frankapy[3] package to send real-time position-control commands to the robot.

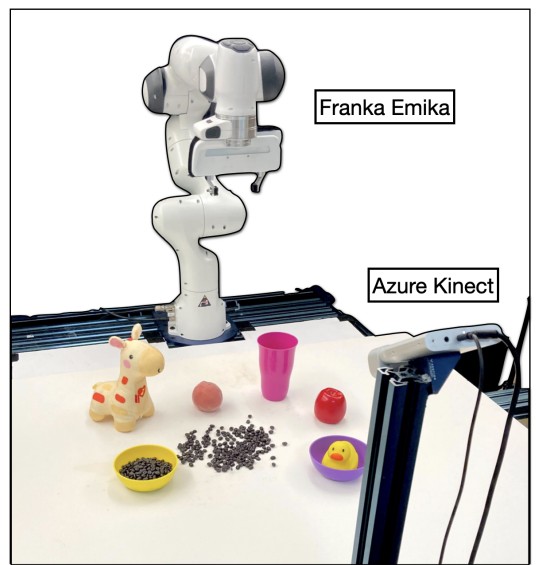

Figure 4: Real-world setup.

## 7.5 Discussion on contribution versus Act3D

The main contribution of Act3D is a lightweight point-based neural architecture that casts keyframe/macro-action prediction as feature point selection with adaptive spatial computation, which brings higher accuracy than previous voxel-based or 2D-based architectures at a lower computation cost. It advances the previous SOTA but still falls under the paradigm of keyframe prediction. ChainedDiffuser, on the other hand, presents a simple yet crucial idea: the unification of the two existing popular paradigms in robotic imitation learning: keyframe abstraction and learnable

---

[1] https://github.com/IFL-CAMP/easy_handeye
[2] https://3dconnexion.com/us/product/spacemouse-compact/
[3] https://github.com/iamlab-cmu/frankapy

Table 4: Success rate comparison with randomized camera viewpoint.

| | pick & lift | pick-up cup | push button | put knife | put money | reach target | slide block | stack wine | take money | take umbrella | Mean | Δ |
|---|---|---|---|---|---|---|---|---|---|---|---|---|
| HiveFormer | 26 | 74 | **98** | 43 | 63 | **98** | 13 | 45 | 77 | 85 | 62.2 | -26.2 |
| ChainedDiffuser | **72** | **91** | 97 | **78** | **88** | 96 | **32** | **90** | **92** | **91** | **82.7** | **-13.1** |

diffusion-based trajectory generation. The two frameworks have been proven successful in the past, but each presents their own drawbacks. ChainedDiffuser is built on top of Act3D, but the way it unifies both frameworks is independent of the macro-action prediction method, and Act3D can be easily swapped with any keyframe prediction approach later if needed. The detailed pipeline proposed and evaluated in our paper is a realization of this unification idea, and proves its effectiveness as a single method that's applicable to a wider range of robotic manipulation tasks.

## 7.6 Robustness of 2D and 3D Methods under Varying Camera Viewpoints

In order to better understand how much it helps to use 3D information in ChainedDiffuser compared with prior 2D methods, we conducted experiments to evaluate the robustness of ChainedDiffuser, InstructRL and HiveFormer under varying camera viewpoint. At test time, we randomly perturb the two shoulder cameras by [20, 30] degrees, while keeping the same camera look-at point. InstructRL completely fails on all the tasks and yields unreasonable actions, as it operates purely on 2D inputs and directly regresses action outputs. It fails to handle distribution shift in image input due to changing camera viewpoint. HiveFormer uses depth image and 2.5D architecture: it selects the highest-score pixel and uses its corresponding depth to compute 3D actions, thus presenting certain robustness. We report numbers of HiveFormer and ChainedDiffuser in Table 6. Our model still performs reasonably well when the cameras are perturbed, achieving the smallest performance drop, showing desirable viewpoint-invariance. This is an advantage of reasoning directly in 3D.

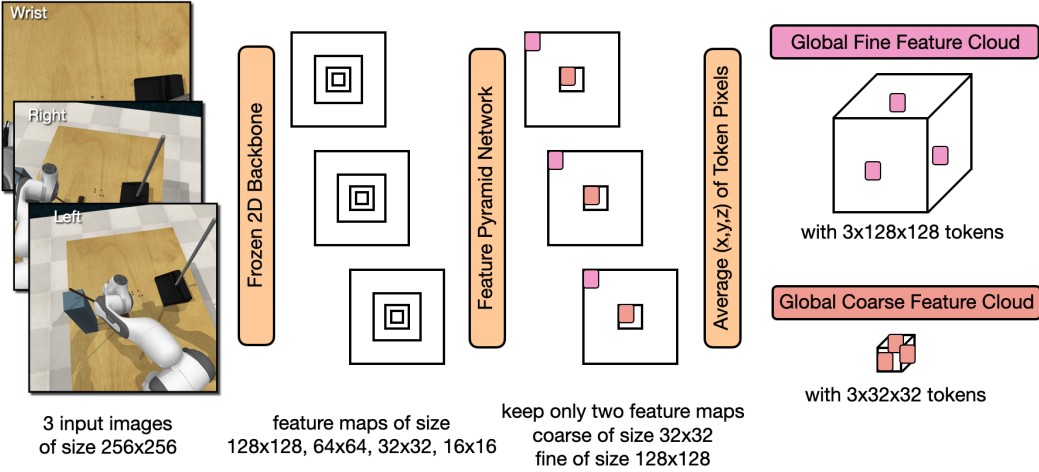

Figure 5: **Scene Feature Cloud Generation**. We encode each image independently with a pretrained and frozen 2D vision backbone to get multi-scale feature maps, pass these feature maps through a feature pyramid network and retain only two: a coarse feature map (at a granularity that lets ghost points attend to all tokens within GPU memory) and a fine feature map (as spatially precise as afforded by input images and the backbone). We lift visual tokens from these two feature maps for each image to 3D scene feature clouds by averaging the 3D positions of pixels in each 2D visual token.

## 7.7 Further Architecture Details

We explain here the our 3D visual backbone in more details, including the visual and language encoding, 3D projection, and relative spatial encoding.

**Visual and language encoder** Our visual encoder maps multi-view RGB-D images into a multi-scale 3D scene feature cloud. We use a large-scale pre-trained 2D feature extractor followed by a feature pyramid network to extract multi-scale visual tokens for each camera view. Our input is RGB-D, so each pixel is associated with a depth value. After featurizing the image, we obtain a feature map whose spatial resolution is lower than the original image. We associate every super-pixel (2D grid location) in this feature map to a depth value, by averaging the depth values of the image pixels that correspond to this super-pixel, i.e., the receptive field. Then we "lift" this 2D feature vector to 3D using the pinhole camera equation and the camera intrinsics, as shown in Figure 5. Each visual token in 3D uses the mean 3D position of all the 2.D pixels as its 3D position.

The language encoder featurizes instructions with a large-scale pre-trained language encoder. We use the CLIP ResNet50 [73] visual encoder and language encoders to exploit their common vision-language feature space for interpreting instructions and referential grounding. Our pre-trained visual and language encoders are frozen during training.

**Relative 3D cross-attentions** ChainedDiffuser uses relative 3D positional encodings proposed in [78, 79] to incorporate translational invariance. We featurize each of the 3D point candidates and a parametric query (used to select via inner-product one of the point candidate as the next best end-effector position in the decoder) independently through cross-attentions to the multi-scale 3D scene feature cloud, language tokens, and proprioception. Our cross-attentions use relative 3D position information and are implemented efficiently with rotary positional embeddings [79]. Given a point $\mathbf{p} = (x, y, z) \in R^3$ and its feature $\mathbf{x} \in R^d$, the rotary position encoding function $\mathbf{PE}$ is defined as:

$$\mathbf{PE}(\mathbf{p}, \mathbf{x}) = \mathbf{M}(\mathbf{p})\mathbf{x} = \begin{bmatrix} \mathbf{M}_1 & & \\ & \ddots & \\ & & \mathbf{M}_{d/6} \end{bmatrix} \mathbf{x}, \ \mathbf{M}_k = \begin{bmatrix} \cos x\theta_k & -\sin x\theta_k & 0 & 0 & 0 & 0 \\ \sin x\theta_k & \cos x\theta_k & 0 & 0 & 0 & 0 \\ 0 & 0 & \cos y\theta_k & -\sin y\theta_k & 0 & 0 \\ 0 & 0 & \sin y\theta_k & \cos y\theta_k & 0 & 0 \\ 0 & 0 & 0 & 0 & \cos z\theta_k & -\sin z\theta_k \\ 0 & 0 & 0 & 0 & \sin z\theta_k & \cos z\theta_k \end{bmatrix}$$

where $\theta_k = \frac{1}{10000^{6(k-1)/d}}$, $k \in \{1, .., d/6\}$. The dot product of two positionally encoded features is

$$\mathbf{PE}(\mathbf{p}_i, \mathbf{x}_i)^T \mathbf{PE}(\mathbf{p}_j, \mathbf{x}_j) = \mathbf{x}_i^T \mathbf{M}(\mathbf{p}_i)^T \mathbf{M}(\mathbf{p}_j)\mathbf{x}_j = \mathbf{x}_i^T \mathbf{M}(\mathbf{p}_j - \mathbf{p}_i)\mathbf{x}_j$$

which depends only on the relative positions of points $\mathbf{p}_i$ and $\mathbf{p}_j$.

