# OpenReview forum: "ChainedDiffuser: Unifying Trajectory Diffusion and Keypose Prediction for Robotic Manipulation"
_robot-learning.org/CoRL/2023/Conference — CoRL 2023 Poster_

### Official Review · Reviewer_JWec · 2023-07-15

**Confidence:** 4
**Originality:** Good
**Technical Quality:** Good
**Clarity Of Presentation:** Fair
**Impact:** 3

**Recommendation:**

Weak Reject: I recommend rejecting the paper, but will not argue for my recommendation if the majority of other reviewers have a different opinion.

**Review:**

**Strengths**

The paper presents convincing empirical results that show the benefits of the proposed approach and clear improvements over baselines.

**Weaknesses**

Summary: The paper could benefit from some more analysis and discussion on when the proposed method is better than alternatives. The writing also needs some major improvement to improve the clarity of the method, and some assumptions could be more clearly stated.

The paper could benefit from some more analysis and discussion on when the proposed method is better than alternatives:

- From Tables 1 and 2, InstructRL also appears to be a strong baseline, and it does not use 3D information. While Act3D is also a strong baseline, it could be seen as an ablation, since several parts of the architecture are shared with this method. It would be great to have some further comparison and analysis against InstructRL in order to understand situations in which (1) the 3D information helps and (2) the overall method performs better than InstructRL. It might also be useful to add InstructRL as a row in Table 4.

- It would also be nice to understand whether Act3D is critical, or if other high-level action detector architectures could be used, especially because Act3D is a concurrent CoRL submission and appears to advance the state of the art itself. This would help decouple the value of this hierarchical policy architecture from the specific instantiation of the high-level.

- The comparison against "Regression" in Table 4 indeed shows the value of using the diffusion model for trajectory prediction. However, the gap is quite small on most tasks and large on a select handful. More analysis of what task characteristics lend themselves to modeling the low-level trajectory generation with diffusion would be valuable.

The writing also needs some major improvement to improve the clarity of the method:

- Section 3.2 is not very clearly explained, and the diagram in Fig 1e is also not straightforward. For example, proprioception is feeding into the cross attention from the bottom of the diagram, while the other inputs feed in from the top. In the text, it isn't clear what Z_query (tilde) and Z_point (tilde) correspond to, and where dot product comparisons are taking place. Instead, having a clear set of equations and variables that describe how inputs map to outputs, instead of pure text descriptions, would be a lot more straightforward to follow.

- Similarly, section 3.3 could also be more clear. For example, the text above equation (2) suggests both cross-attention and self-attention operations are being applied simulataneously, which does not make much sense, and equation (3) has the same variable appearing three times. Perhaps equation (2) and (3) are describing the evolution of a particular variable (e.g. a forward pass through some operations) but it is not clear right now.

- More details on the way 2D CLIP vision tokens are unprojected into 3D would be helpful. The point-based representation appears to be a crucial part of the method, so clear and precise operations (with equations) would be helpful (at the very least in the appendix).

- Overall, the writing in section 3 could be improved by deferring some details (such as relative positional encodings) to the appendix, and instead focusing more on the core operations that describe how the input is transformed into the output prediction.

- Improving the figures would also be beneficial. Consider adding more figures of the model architecture and splitting the dense overview figure (Figure 1) into more pieces. Some of the results figures (e.g. Tables) could be compressed and/or moved to the appendix to make room for more figures that outline the approach more clearly, and provide intuition.

- The method requires a calibrated camera - this should probably be stated clearly as a limitation / assumption in the paper. It might also be useful to explicitly state the assumptions on the data contained in the demonstrations.

Some additional general comments follow:

- L2 loss for quaternion prediction (as mentioned above equation 5) seems like it could be problematic. How is the network output enforced to be a valid quaternion?

- Reference 6 and 10 are repeated.

**Quality Of The Limitations Section:**

Limitations are addressed clearly

**Questions For Rebuttal:**

Please address the points made in the weaknesses section -- in particular the comments on experimental evaluation and clarity.

**Robotics Focus:**

Sufficient demonstration on hardware

**Summary Of Paper:**

This paper presents ChainedDiffuser, a new policy architecture for learning language-conditioned robot manipulation tasks via imitation learning. The architecture predicts high-level actions (e.g. a next end effector pose for the arm to reach) based on RGBD input and language conditioning and uses a diffusion model, conditioned on the high-level action, to generate a dense trajectory to reach it. The experiments demonstrate strong performance on the RLBench benchmark, and on some real world tasks. A comparison to ablations also highlights the value of both the high-level and low-level components.

**Summary Of Recommendation:**

While the experimental results in the paper appear to be strong, the paper would benefit from some additional discussion and comparisons. The paper should also revise the writing to improve the clarity.

---

### Official Review · Reviewer_BFkn · 2023-07-16

**Confidence:** 4
**Originality:** Good
**Technical Quality:** Excellent
**Clarity Of Presentation:** Very Good
**Impact:** 3

**Recommendation:**

Weak Accept: I recommend accepting the paper, but will not argue for my recommendation if the majority of other reviewers have a different opinion.

**Review:**

The paper is overall well explained, the work is of high technical quality and significant importance. The method is reasonably novel.

The main strength of the paper is that it overcomes the limitations of methods based only on transformers or diffusion by cleverly combining them, achieving state-of-the-art on RLBench, significantly improving performance on contact-rich tasks. The method is also very well justified and the experiments are sound.

The fact that their concurrent work Act3D performs competitively in most benchmarks to ChainDiffusers, and is submitted at the same time begs the question of whether the two should have been submitted together, with Act3D simply being a baseline to further show the improvements and add to the novelty of ChainDiffusers.

Some of the weaknesses are well highlighted in the limitations section, the most significant one being that the method performs closed-loop control on the macro-actions, which could make it brittle in dynamic environment.

**Quality Of The Limitations Section:**

Limitations are addressed clearly

**Questions For Rebuttal:**

One remaining question is about the extent of the novelty presented in unifying two frameworks in itself,  regardless of how good the resulting method and technical contributions are. It would be appreciated to see a more clear statement of the author's view of this.

Moreover, as pointed out already in the limitations, ChainedDiffuser is not very flexible in highly dynamic environments, but since it's mentioned it would have been sound to see some experiments on this for completeness.



**Robotics Focus:**

Sufficient demonstration on hardware

**Summary Of Paper:**

The paper proposes ChainedDiffuser, unifying transformer-based macro action prediction and diffusion-based trajectory generation, allowing it to leverage their respective strengths. Namely transformer-based macro action prediction are good at long horizon planning but struggle with continuous interactions and smooth trajectory generation; diffuser-based trajectory generations perform well in local smooth trajectory generation but struggle with long horizon multi-task learning. ChainedDiffuser thus uses a transformer-based macro-action prediction, and creates smooth micro-actions to fill in the gap between macro-actions with a diffusion-based trajectory generator.

They test their method on RLBench, and demonstrate state-of-the-art performance in a variety of challenging settings. They show that ChainedDiffuser outperforms transformer-based only methods and trajectory diffusion alone, justifying the unification of the two frameworks.

**Summary Of Recommendation:**

The paper is technically sound, well written and potentially of high impact. Its contributions are somewhat novel. The claimed results are better than previous state-of-the-art. The recommendation is weak accept, but could be upgraded to a strong accept if the authors point out more clearly if and how significant the novelty is, despite the great technical quality of the paper.

---

### Official Review · Reviewer_zryn · 2023-07-20

**Confidence:** 4
**Originality:** Good
**Technical Quality:** Very Good
**Clarity Of Presentation:** Very Good
**Impact:** 4

**Recommendation:**

Weak Accept: I recommend accepting the paper, but will not argue for my recommendation if the majority of other reviewers have a different opinion.

**Review:**

Act3D proposes most of the segment in the proposed approach, from obtaining the language task and 3D feature cloud embeddings to getting the goal state which leads to the successful completion of the task. This is also evident from the success metric values of Act3D that a simple planner is sufficient for most tasks.

When that is not the case, the authors show that a learning-based planner is beneficial and show that the performance of a diffusion-based planner is better as compared to a regression-based action sequence fitting (also established by prior work: Diffusion Policy).

If we consider Act3D as prior work, then the proposed method has limited novelty. This is more concerning as the architecture for obtaining all the used embeddings is also similar to Act3D. However, the unified framework is an efficient approach for solving complex manipulation problems in 3D. The span of environments used for validation of the proposed approach is convincing. Further, the ablation on the importance of the unified framework as compared to using individual segments justifies the combination.

**Quality Of The Limitations Section:**

Limitations are addressed clearly

**Questions For Rebuttal:**

1. How do you actually achieve this “We train both the action detector and the trajectory diffuser jointly” using Eq. (5)? Aren’t you using score matching objective for training diffusion score network? Eq (5) seems to be the training objective for regression planner.

2. The limitation mentions that the control is closed-loop w.r.t. the macro-actions (i.e. goal). This is confusing as no clarifications on control is provided in the implementation section. Are the micro-actions open-loop for 50 poses? Please add an algorithm to describe the control loop.

3. In Table 1, is chained-diffuser not using a learned collision avoidance classifier for the low-level motion planner?

4. Are the language instructions same for a particular task? How do they change during test time?

**Robotics Focus:**

Sufficient demonstration on hardware

**Summary Of Paper:**

The presented paper introduces a hierarchical method to solve language-conditioned tasks. For a given natural language-based task description and multi-view images of the scene, a 3D feature cloud is constructed and used to condition a macro-action policy to find the goal action (end-effector pose). This segment is from Act3D, their concurrent submission. Next, the goal state and the current state, along with the scene and task embeddings, are used to condition a trajectory diffuser to sample intermediate micro-actions for obtaining the whole sequence of poses (contribution). The method proposes goal pose (Diffuser) to action-only diffusion (Diffusion Policy) for fixed number of intermediate poses between current and goal state. The proposed method is validated over RLBench tasks in simulation and real-world deployments. Comparison is conducted with relevant baselines in their corresponding benchmarks. Ablation results are shown for some tasks to show the relevance of each component.

**Summary Of Recommendation:**

The paper is built over Act3D, where a classical motion planner between the current and predicted goal state is replaced by a diffusion-based planner. While such a framework helps marginally in most of the tasks, certain challenging tasks have been solved well by the unified approach. There is minor discrepancy in how the authors are training the diffusion model jointly with their macro-action model and how the control loop is working. The evaluation is very extensive and appropriate comparison results have been shown.

---

### Author Response · Authors · 2023-08-14
**General Response**

We thank all the reviewers for their detailed feedback and useful suggestions! The reviewers found the proposed method "efficient in addressing complex manipulation problems" (zryn), "well explained and presents high technical quality and significant importance" (BFkn), with "valuable components" (JWec) and  with "extensive and convincing empirical results" (zryn, JWec). We address all the concerns of the reviewers in the individual responses.

We have also updated our manuscript with requested revisions. We summarize the major changes below:

- We updated our method section (3.2, 3.3 & 3.4), with clearer equations, clearer explanations and additional details of our policy architecture, its training and inference. [zryn, BFkn, JWec]
- We included additional comparisons with InstructRL in more settings (Tables 1,2 and 4 in the updated paper), as well as experiments where we vary the camera placement at test time  to demonstate the benefit of   reasoning in 3D over 2D manipulation policies, such as InstructRL, in Section 6.6 of the Appendix. [JWec]
- We added one dedicated section (Appendix Section 6.5) discussing our perspective on how ChainedDiffuser differs from Act3D and the major contribution contained in each paper [zryn, BFkn]
- We added more architecture details on the visual and language feature encoding, 3D computation, and relative positional encoding in Appendix Section 6.7 [JWec]


We hope our responses address all reviewers' concerns convincingly. We would like to express our gratitude towards all the reviewers again for their detailed and constructive feedback.

---

### Decision · Program_Chairs · 2023-08-30

**Decision:**

Accept (Poster)

**Comment:**

This paper presents ChainedDiffuser, a new policy architecture for learning language-conditioned robot manipulation tasks.  The proposed method overcomes the limitations of methods based only on transformers or diffusion by cleverly combining them, achieving state-of-the-art on RLBench. Reviewers agree that the method is well justified, clearly described, and the experiments are sound. It is an important contribution to the field. However, the reviewers also raise some concerns about the comparison with concurrent work such as Act3D. Since it is a concurrent work, the AC thinks a comparison is not required for submission, but it will be nice to include a discussion and potential comparison in the final version.